# Effects of Probiotic *Lactiplantibacillus plantarum* HJLP-1 on Growth Performance, Selected Antioxidant Capacity, Immune Function Indices in the Serum, and Cecal Microbiota in Broiler Chicken

**DOI:** 10.3390/ani14050668

**Published:** 2024-02-21

**Authors:** Caimei Yang, Shuting Wang, Qing Li, Ruiqiang Zhang, Yinglei Xu, Jie Feng

**Affiliations:** 1Key Laboratory of Applied Technology on Green-Eco-Healthy Animal Husbandry of Zhejiang Province, College of Animal Science and Technology, Zhejiang Agriculture & Forestry University, Hangzhou 311300, China; yangcaimei2012@163.com (C.Y.); 13730762487@163.com (S.W.); 18806516157@163.com (Q.L.); zrq1034@163.com (R.Z.); 2Key Laboratory of Animal Feed and Nutrition of Zhejiang Province, College of Animal Sciences, Zhejiang University, Hangzhou 310058, China; fengj@zju.edu.cn

**Keywords:** *Lactiplantibacillus plantarum* HJP-1, broiler, growth, serum biochemical parameters, cecal microbiota

## Abstract

**Simple Summary:**

With the ban on antibiotics being added to feed as growth promoters, numerous probiotics have been developed as an alternative to antibiotics that are added to rations. Although a considerable number of studies have shown that *Lactiplantibacillus plantarum* (*L. plantarum*) has positive effects on intestinal health, more evidence is needed to establish whether *L. plantarum* plays an efficient role in replacing antibiotics as a growth promoter in broiler nutrition. Therefore, the present work evaluated the effects of *L. plantarum* on growth performance, selected antioxidant capacity, and immune function indices in the serum, the microbiota, and some short-chain fatty acids in the cecum of broilers and compared them with those of antibiotic growth promoters.

**Abstract:**

This research study aimed to investigate the effects of *Lactiplantibacillus plantarum* (*L. plantarum*) on growth performance, oxidation resistance, immunity, and cecal microbiota in broilers. This work classed three hundred and sixty 1-day-old male broilers into three groups randomly, including a control group (CON, basal diet) and antibiotic (ANT, 75 mg kg^−1^ chlortetracycline added into basal diet) and probiotic groups (LP, 5 × 10^8^ CFU kg^−1^
*Lactiplantibacillus plantarum* HJLP-1 contained within basal diet). Animals were then fed for 42 days, and each group comprised eight replicates with 15 broilers. Compared with CON, *L. plantarum* supplementation significantly improved the average daily weight gain (AWDG) (*p* < 0.05) while reducing the feed–gain ratio over the entire supplemental period (*p* < 0.05). Birds fed *L. plantarum* had markedly lower serum ammonia and xanthine oxidase levels (*p* < 0.05) than those in the ANT and CON groups. Significant improvements (*p* < 0.05) in superoxide dismutase, catalase, and serum IgM and IgY contents in broilers fed *L. plantarum* were also observed when compared with those in the CON and ANT groups. Both *L. plantarum* and antibiotics decreased pro-inflammatory factor IL-1β levels significantly (*p* < 0.05), while only *L. plantarum* promoted anti-inflammatory factor IL-10 levels in the serum (*p* < 0.05) compared with CON. *L. plantarum* (*p* < 0.05) increased acetic acid and butyric acid concentrations in cecal contents when compared to those in CON and ANT. Among the differences revealed via 16S rRNA analysis, *L. plantarum* markedly improved the community richness of the cecal microbiota. At the genus level, the butyric acid-producing bacteria *Ruminococcus* and *Lachnospiraceae* were found in higher relative abundance in samples of *L. plantarum*-treated birds. In conclusion, dietary *L. plantarum* supplementation promoted the growth and health of broilers, likely by inducing a shift in broiler gut microbiota toward short-chain fatty acid (SCFA)-producing bacteria. Therefore, *L. plantarum* has potential as an alternative to antibiotics in poultry breeding.

## 1. Introduction

Broiler chickens are one of the main sources of animal protein for human consumption. To maximize production, intensive indoor systems are employed by commercial poultry producers [1]. However, intensively reared broiler chickens encounter the simultaneous action of multiple stressors from the housing environment, which causes high mortality in broilers from common infectious or metabolic diseases [2]. Conventional poultry farms use antibiotics for treatment, prophylaxis, and growth promotion, which gives rise to antibiotic-resistant pathogens. The emergence and spread of antibiotic resistance affect animal and human health and have become a global concern [3,4]. Therefore, many countries have made efforts to limit antibiotic application in feed. In 2020, antibiotics being added to feed as growth promoters was banned in China.

With the ban on antibiotics, an increasing number of probiotics are being developed as an alternative to antibiotics that are added to diets [5,6]. Suitable probiotics can improve intestinal microbiota, enhance the absorption of nutrients by livestock and poultry, strengthen immunity, and reduce the impact on farm animal health and growth induced by the feeding environment [7].

*Lactiplantibacillus plantarum* is a homofermentative, anaerobic–aerotolerant, Gram-positive bacterium that produces both isomers of lactic acid as the main end products of carbohydrate fermentation [8]. It has been commonly used as a probiotic due to its outstanding probiotic qualities. Some researchers have demonstrated that different strains of *LactIplantibacillus plantarum* have good gastrointestinal tolerance and good adhesion [9,10,11]. Yang et al. (2019) observed the antioxidant and immune-enhancing effects of the probiotic *Lactobacillus plantarum*, isolating 200,655 from kimchi [12]. Furthermore, some research has revealed that the fermentation metabolism of *L. plantarum* produces some organic acids, such as lactic, citric, isobutyric, and acetic acids; ethanol; diacetyl; and H_2_O_2_, which result in a decrease in the pH of the medium, inhibiting the development of the pathogen [9,10,13]. In broiler chickens, feeding *L. plantarum* can improve growth performance, promote gut microbial homeostasis, prevent pathogen infection, and reduce the intestinal barrier injury caused by toxins [14,15,16]. In light of the information presented above, we evaluated *L. plantarum* to modulate growth and health in broiler chickens compared with antibiotic growth promoters to establish whether *L. plantarum* plays an efficient role in replacing antibiotics as a growth promoter in broiler feed. To complete the objective mentioned, in addition to growth performance, several parameters were studied, such as the antioxidant concentration, immune function indices, cecal microbiota, and some short-chain fatty acids, as these are the key factors to determine the health status of these birds.

## 2. Material and Methods

### 2.1. Experimental Animals and Design

A total of 360 one-day-old male Avian broilers were individually weighed and randomly assigned into three groups: (1) a control group (CON), fed a basal diet; (2) an antibiotic group (ANT), fed the base diet supplemented with 75 mg/kg of chlortetracycline; (3) a probiotic group (LP), fed the basal diet plus 5.03 × 10^8^ CFU/kg of *L. plantarum* HJLP-1. The preparation and quality inspection of *Lactiplantibacillus plantarum* HJLP-1 powder were completed by Vegamax Biotechnology Co., Ltd. (Huzhou, China). The number of viable bacteria in the powder was 5.03 × 10^8^ CFU/g, and the bacterial powder addition in the broiler feed was 1000 mg/kg. Each treatment had 15 birds of 8 replicates in each cage (110 cm × 90 cm × 40 cm). Dietary formulations met or exceeded the daily growth needs of broiler chickens (NRC, 1994; Table 1). The feeding experiment lasted for 42 d. Each cage was equipped with a hanging feeder and nipple drinker water line. Animals had free access to water and mashed feed. The room temperature in the first week was set at 35 °C and was gradually decreased to 25 °C by the end of the experiments. The lighting condition followed the standard of 23 h of light and 1 h of darkness. The relative humidity in the rearing room was maintained at 45% to 55%. Immunization, disease prevention, and disinfection were carried out via routine methods.

### 2.2. Sample Collection

The weights of broilers from each pen were taken on days 1, 21, and 42 to evaluate the average weight daily gain (AWDG). Meanwhile, the remaining feed was weighed to calculate the average daily feed intake (ADFI) together with the feed–gain ratio (F/G). 

One bird from each replicate pen was randomly chosen to collect samples on day 42 (eight broilers from every group). Left-wing venous blood collection was completed using 5 mL sterilized tubes, and the samples were centrifugated at 3000× *g* and 4 °C for a 10 min period (Centrifuge 5424R, Eppendorf, Hamburg, Germany) to obtain serum. After blood collection, the broilers were sacrificed. The cecal content of each broiler was gently squeezed into sterile cryotubes, snap-frozen on dry ice, and stored at −80 °C for microbial genomic DNA extraction. Between the sampling of each broiler, sterile gloves were changed, and the table, scissors, and tweezers were cleaned with 70% ethanol to prevent cross-contamination between samples.

### 2.3. Measurements of Biochemical Parameters in Serum

Serum samples were screened for biochemical parameters with respect to metabolites, including ammonia (NH_3_, Cat. YH1312); urea nitrogen (UN, Cat. SD2260); uric acid (UA, Cat.YH1261) concentrations; and antioxidants, including xanthine oxidase (XOD, Cat. YH1219). Malondialdehyde (MDA, Cat.YH1217), superoxide dismutase (SOD, Cat.YH1200), catalase (CAT, Cat.YH1206), and glutathione peroxidase (GSH-Px, Cat.YH1267) activities were measured with commercial assay kits (Nanjing Angle Gene Bioengineering Co., Ltd. Nanjing, China), and the measurements were performed using an automatic biochemical analyzer (Toshiba, Tokyo, Japan). The concentrations of immunoglobulins (IgA, Cat. ANG-E32004C; IgM, Cat. ANG-E32005C; IgY, Cat. ANG-E32209C) and the cytokines of interleukin-1β (IL-1β, Cat. ANG-E32031C), interleukin-6 (IL-6, Cat. ANG-E32013C), interleukin-10 (IL-10, Cat. ANG-E32011C), tumor necrosis factor α (TNF-α, Cat. ANG-E32030C), and interferon-β (IFN-β, Cat. ANG-E32002C) were determined using enzyme-linked immunoassay (ELISA) analysis kits obtained from Nanjing Angle Gene Bioengineering Co., Ltd. (Nanjing, China). According to the manufacturer’s protocol, standard solutions or serum samples were added to 96-well plates (coated with purified chicken: IL-1β, IL-6, IL-10, TNF-α, and IFN-β antibodies). Then, a second horseradish peroxidase-labeled antibody was added to the wells, and the plates were incubated at 37 °C for 1 h. After washing the wells 5 times, chromogen solutions were added and preserved in the dark for 15 min at 37 °C. Finally, absorbance was measured at 450 nm using a multifunctional microplate reader (Tecan Infinite M200 PRO; Grödig, Austria) after the addition of the stop solution.

### 2.4. Assessment of Cecal Microbiota Contents via 16S rRNA Sequencing

Cecal microbial genomic DNA was extracted using the QIAamp DNA Stool Mini Kit (Qiagen GmbH, Hilden, Germany). The final DNA concentration and purity were determined using a NanoDrop 2000 UV–Vis spectrophotometer (Thermo Scientific, Wilmington, NC, USA), and DNA quality was further verified and monitored via 1% agarose gel electrophoresis. Then, the broad-range PCR amplification of the V3–V4 hypervariable region of the 16S rRNA gene was performed using 338F forward-primer formulation-targeting domain bacteria, along with an 806R reverse primer with 8 bp barcodes to facilitate multiplexing [17], and sequenced on the Illumina MiSeq platform (Illumina, San Diego, CA, USA). Finally, the composition and abundance of the caecum microbiota were determined via alpha and beta diversity analysis through the free online Majorbio Cloud Platform: https://cloud.majorbio.com (accessed on 15 October 2020). The alpha diversity (Chao1 estimates, observed species, Shannon index, abundance-based coverage estimator [ACE], and Simpson indices) was calculated in Mothur [18]. To obtain the beta diversity, weighted UniFrac was calculated using the QIIME2 (1.0) software and subject to principal component analysis (PCA) with the ape package in R (https://cloud.majorbio.com) [19]. Furthermore, linear discriminant analysis coupled with the effect size (LEfSe) algorithm within the Mothur program was used to determine the community that significantly affected the sample’s division [20]. A linear discriminant analysis score threshold of >4.0 was selected as significantly different for CON, ANT, and LP. Venn diagrams were also generated to compare the numbers of common and unique OTUs among three groups using the web-based InteractiVenn tool (https://cloud.majorbio.com). Correlations between SCFA and taxonomic relative abundance at the phylum and genus levels were determined using Spearman correlation coefficients. Spearman’s rank correlations and *p*-values were calculated with the psych package (2.1.6).

### 2.5. Short-Chain Fatty Acid Concentration Analysis

The concentration of cecal short-chain fatty acid (SCFA) was estimated via headspace sampler gas chromatography (Agilent Technologies, Wilmington, DE, USA) using the method of Thanh et al. [21]. The 0.5 g sample was dissolved in 1 mL of water. After centrifugation at 12,500× *g* for 5 min, the supernatant was extracted and mixed with 25% phosphoric acid. The concentration of SCFAs was determined via an Agilent Technologies 7890A Network System equipped with a 30 m × 0.32 mm × 1.8 μm column (DB-624) and flame ionization detector.

### 2.6. Statistical Analysis

One-way analysis of variance (ANOVA) followed by Tukey’s method for comparing differences was performed using GraphPad Prism 8.2.1 software (GraphPad Prism Inc., La Jolla, CA, USA) depending on the normality of variable distributions (verified with a Kolmogorov–Smirnov test) and the homogeneity of variances (verified with the Levene test). Results were expressed as means with the standard error of the mean (SEM). Statistical significance was set at *p* < 0.05. Analyses of bacteria community data were conducted using the cloud majorbio platform (http://cloud.majorbio.com/; accessed on 15 October 2020).

## 3. Results

### 3.1. Growth Performance

Table 2 displays the growth performance analysis. In comparison with the control, both *L. plantarum* HJLP-1 and antibiotic treatments improved the body weight (BW) significantly (*p* < 0.05) on the 21st day. In contrast, on the 42nd day, only the LP group showed remarkably elevated BW (*p* < 0.05). Accordingly, during the starter period (1–21 d), the birds fed *L. plantarum* HJLP-1 or antibiotics had a higher average weight daily gain (AWDG) than the birds in the CON group (*p* < 0.05). However, the average daily feed intake (ADFI) in the LP group was significantly higher (*p* < 0.05) than in the CON group; thus, there was no obvious difference in F/G between the two groups. In the finisher period (22–42 d) and during the entire supplementation period (1–42 d), the LP group had markedly elevated AWDG compared with the ANT and CON groups (*p* < 0.05) without an obviously increased ADFI. Therefore, during the starter period, supplementation with antibiotics decreased the F/G compared with the LP and CON groups (*p* < 0.05). In contrast, in the finisher period and during the entire supplementation period, broilers fed with *L. plantarum* HJLP-1 had a close F/G ratio to those of the ANT group, and the ratio obviously decreased relative to the CON group (*p* < 0.05).

### 3.2. Serum Metabolite Index

The effects of *L. plantarum* HJLP-1 on the metabolite index in broiler serum are shown in Table 3. Although either *L. plantarum* HJLP-1 or antibiotic supplementation significantly decreased the serum ammonia (NH_3_) level (*p* < 0.05) relative to the CON group, the serum ammonia levels of broilers fed *L. plantarum* HJLP-1 dramatically decreased compared with broilers fed with antibiotics (*p* < 0.05). The serum urea nitrogen (UN) and uric acid (UA) contents were not significantly different. Furthermore, birds in the LP group had lower serum xanthine oxidase (XOD) levels than birds in the CON and ANT groups (*p* < 0.05). 

### 3.3. Antioxidant Index in Serum

The effects of *L. plantarum* HJLP-1 on the antioxidant index in serum are shown in Table 4. The addition of *L. plantarum* HJLP-1 significantly improved the SOD and CAT levels in the serum (*p* < 0.05). MDA and GSH-PX contents were not significantly different across diverse groups. 

### 3.4. Immunoglobulin and Cytokines Index in Serum

As shown in Table 5, both *L. plantarum* HJLP-1 additions markedly elevated serum IgM and IgY contents in broilers (*p* < 0.05) compared with that in the CON and ANT groups. Compared with CON, *L. plantarum* HJLP-1 supplementation reduced pro-inflammatory factor IL-1β levels significantly (*p* < 0.05) and promoted anti-inflammatory factor IL-10 expression dramatically (*p* < 0.05). Antibiotics also markedly reduced IL-1β levels (*p* < 0.05), but they had no effect on IL-10 levels. In addition, pro-inflammatory factor TNF-α levels in the serum of broilers fed *L. plantarum* HJLP-1 were remarkably decreased compared with broilers fed antibiotics (*p* < 0.05). 

### 3.5. Microbiota Structure in the Cecal Contents

As shown in Figure 1A–E, *L. plantarum* supplementation induced some changes in microbiota structure in the cecal digesta of broilers. The Venn diagram demonstrated that the unique operational taxonomic units (OTUs) of the LP group were 166, which is substantially higher than those of the CON (39) and ANT (24) groups (Figure 1A). According to genus-relative abundance-based principal component analysis (PCA), the microbiota in the LP group were clearly separated from the CON and ANT groups (Figure 1B). Furthermore, *L. plantarum* supplementation markedly improved community richness, as evidenced by the significantly increased ACE and Chao indices, compared to other groups (*p* < 0.05) (Figure 1C). Samples from the ANT and LP groups had higher proportions of *Firmicutes* and a lower ratio in *Bacteroidetes* than those from the CON group at the phylum level (Figure 1D). The results of linear discrimination analysis coupled with the effect size (LEfSe) indicated that *Lactoplantibacillus plantarum* supplementation mainly enriched *f*_*Ruminococcaceae* and *unclassified* _*f*_*Lachnospiraceae;* antibiotic supplementation enriched *Alistipes* and *Oscillospirales*, whereas *Bacteroides, Bacteroidaceae*, and *Barnesiellaceae* were enriched at the genus level of control broilers (Figure 1E).

### 3.6. Short-Chain Fatty Acid Concentrations

Figure 2 displays SCFA results. Butyric acid and acetic acid within the cecal contents of broilers fed *L. plantarum* HJLP1 dramatically increased (*p* < 0.05) compared with ANT and CON. There was no significant modification in propionic acid concentrations among the three treatment groups.

### 3.7. Link between Cecal Bacterium Community and Short-Chain Fatty Acids

Figure 3 shows the Spearman correlation heatmap of the top 50 genera and short-chain fatty acids. During digestion, *f*_*Ruminococcaceae, Alistipes,* and *unclassified_f_Lachnospiraceae* were significantly positively correlated with butyric acid production, whereas *Bacteroides* had a significant negative correlation.

## 4. Discussion

Probiotics are widely reported to promote broiler health and development [22,23,24]. *Lactoplantibacillus plantarum* has also been shown to promote growth in broilers [25,26,27]. Consistent with these previous reports, chickens fed *L. plantarum* HJLP-1 exhibited higher BW and AWDG than chickens fed only a basal diet. During the starter period, ADFI increased in the LP group, indicating that *L. plantarum* HJLP-1 may improve the AWDG by promoting the chicken’s appetite. In the finisher period, *L. plantarum* HJLP-1 supplementation increased AWDG without influencing feed intake (FI), resulting in a reduced F/G ratio (improved feed conversion) over the entire supplemental period to a level that was equivalent to or lower than that of antibiotics. Therefore, the increased AWDG might be attributed to the improvement in nutrient utilization. These beneficial effects on growth performance indicate that *L. plantarum* HJLP-1 could be an effective substitute for antibiotics in poultry diets. 

Serum uric acid (UA) and urea nitrogen (UN) have been considered as indicators for evaluating the utilization of amino acid in broilers [28]. Ammonia is inevitably liberated in protein metabolism, but it is very toxic relative to living cells, and the sensitive mechanisms in animals keep it below a toxic level. According to some research studies, there is a strong correlation between serum ammonia levels and the capacity to utilize all digestible plasma-free essential amino acids in protein synthesis in broiler chicks. Furthermore, the blood ammonia level also is an important factor in the regulation of appetite [29]. Elevated serum indexes can cause disruptions in amino acid and nucleotide metabolism, ultimately affecting antioxidant capacity [30] and the immune function of the body [31]. In this study, *L. plantarum* HJLP-1 supplementation reduced serum UA, UN, ammonia (*p* < 0.05), and XOD (*p* < 0.05), compared with CON and ANT. Hence, *L. plantarum* HJLP-1 may have a positive effect on amino acid utilization and appetite in broiler diets. XOD represents a critical enzyme that is responsible for UA production. Enhanced XOD activity can assist in generating UA by producing reactive oxygen species (ROS) [32]. 

Malondialdehyde, T-SOD, CAT, and GSH-Px represent major factors adopted for assessing the oxidative status of the enzymatic system. Due to lipid peroxidation, produced MDAs can damage physical cell activity and alter biological membrane function, which has been frequently adopted to be a biomarker for oxidative injury. In contrast, SOD protects cells from free radicals by catalyzing the conversion of superoxide into oxygen and oxygen peroxide (H_2_O_2_). Later, CAT and GSH-PX can convert the obtained hydrogen peroxide to water to protect cells against damage due to oxygen stress [33]. In vitro studies revealed that *L. plantarum* fermentation potently reduces and scavenges free radicals [34,35]. In vivo studies have shown that *L. *plantarum** colonizes the intestinal tract, which has an important effect on protection against free radicals. Li et al. (2021) [36] found that feeding *Lactoplantibacillus plantarum* KSFY06 at high concentrations can inhibit the decrease in oxidation-associated enzymes and complexes in serum and liver caused by liver injury effectively. Izuddin et al. (2020) [37] reported that a diet with *Lactoplantibacillus plantarum* supplementation in the diet of post-weaning lambs can improve antioxidant levels in serum and rumen and upregulate antioxidase activities in the liver and rumen barrier function. Similarly to these studies, adding *L. plantarum* HJLP-1 to the diet significantly increased serum T-SOD and CAT activities in broilers. Therefore, *L. plantarum* HJLP-1 promoted the antioxidant capacity of broilers mainly by scavenging free radicals. 

Serum immunoglobulin, a key indicator of an animal’s humoral immunity status, represents a defense mechanism against the intrusion of foreign substances into the living body. IgA, IgG (IgY), and IgM represent the critical immunoglobulins of avian species [38,39]. According to our results, supplementation with *L. plantarum* HJLP-1 enhanced broilers’ immunity by promoting serum IgM and IgY levels. Such results conform with other studies [40], where feeding a postbiotic produced by *L. plantarum* improved the serum IgM and IgY levels in broilers. Based on the fact that cytokines have essential effects on inflammatory and immune responses, their balance exerts a critical effect on resisting infection. Increases in pro-inflammatory factors, like TNF-α, IL-6, and IL-1β, induce systemic inflammation reaction and tissue damage [41], while IL-10 is the anti-inflammatory and tolerogenic factor blocking pro-inflammatory factor production [42]. Some studies have reported that *L. plantarum* could upregulate anti-inflammatory factors while suppressing pro-inflammatory factors within the intestinal mucus of broilers challenged by toxins or pathogens [15,43]. The results of this study revealed that *L. plantarum* HJLP-1 also decreased serum IL-1β expression while increasing IL-10 expression, which indicated that supplementation with *L. plantarum* HJLP-1 might be beneficial for attenuating the systemic inflammatory response in broilers. 

Numerous studies have concluded that probiotics have positive effects on the health, performance, and disease resistance of hosts, as they can regulate the gut microbial flora balance in hosts by inhibiting the proliferation of pathogenic species and improving the number of beneficial bacteria [35,44,45]. Therefore, the beneficial effects of *L. plantarum* HJLP-1 on broilers in this study may be attributed to its influence on the endogenous commensal microbiota and changes in metabolite production induced by intestinal bacteria. 

SCFAs, especially butyric acid, propionate, and acetic acid, represent the main bacterial fermentation end products of dietary components, especially indigestible fibers in the intestine, and they play important roles in maintaining the health of the host. Both acetic acid and lactic acid were detected in postbiotics produced by a variety of *Lactobacillus plantarum* [46]. Acetic acid can contribute to environmental acidification and inhibit the growth of less acidophilic organisms (most pathogenic bacterial species) in the same ecosystem [47]. Butyric acid has been shown to have a critical effect on growth promotion [48,49], oxidative stress reduction [50], immune modulation, anti-inflammatory effects [51], and the inhibition of colonization by pathogenic bacteria in broilers [52]. According to our results, the dramatically increased butyric acid and acetic acid concentrations in the cecal contents of broilers fed *L. plantarum* HJLP-1 may be partly responsible for the beneficial effects of *L. plantarum* HJLP-1 introduced to broilers.

Diverse microorganisms densely colonized in the gastrointestinal tract of poultry have been considered to have vital effects on host health and growth performance [53]. Our results indicated that *L. plantarum* HJLP-1 plays a role in modulating the gut microbiota of broilers. It can enrich the cecal microbiota communities with unique microbiota, as reflected by significantly increased ACE, Chao richness indices, and abundance of unique OTUs in LP compared to CON and ANT. Some researchers indicated that gut microbiota modulation by *L. plantarum* was associated with beneficial effects since *L. plantarum* favored the growth of the Firmicute phylum, which includes butyric acid-producing bacteria [54,55]. Consistent with these research studies, in our research, compared to the CON group, increased Firmicute abundances with simultaneously decreased Bacteroidete abundances in the LP and ANT groups were observed, considering an increase in the Firmicutes/Bacteroidetes ratio at the phylum level, which showed positive relations to an increase in BW in broilers [56,57]. This may be a possible reason for the growth-promoting effects of *L. plantarum* HJLP-1 and antibiotics. Moreover, as indicated by the Spearman correlation heatmap, *Ruminococcaceae* and *Lachnospiraceae*, which were enriched in the LP group, promoted butyric acid production. *Alistipes*, which are enriched in the ANT group, also had a positive correlation with butyric acid production, while *Bacteroides*, which were enriched in the control group, inhibited butyric acid production. This is probably the main reason for the increase in butyric acid by *L. plantarum* HJLP-1 and antibiotics. 

## 5. Conclusions

Taken together, *Lactoplantibacillus plantarum* HJLP-1supplementation could improve growth performance and promote the antioxidant capacity and immune function of broilers. These beneficial effects of *L. plantarum* HJLP-1 on broilers may be attributed to its influence on some specific bacterial abundance in the intestinal microbiota, resulting in a marked increase in butyric acid and acetic acid contents. Therefore, *L. plantarum* HJLP-1 could be an efficient substitute for antibiotics in promoting the growth and health of broilers during production.

## Figures and Tables

**Figure 1 animals-14-00668-f001:**
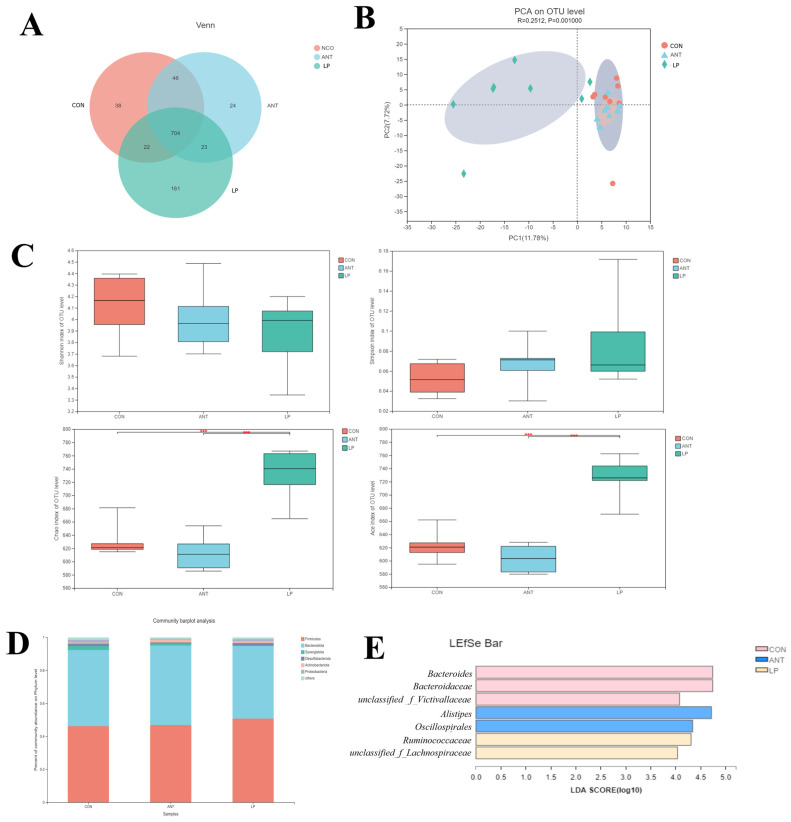
Analysis of the composition of fecal microbiota. (**A**) Venn diagram summarizing the numbers of common and unique observed taxonomic units (OTUs) in the microbiota community in the cecum contents of broiler chicken. (**B**) Principal component analysis (PCA). (**C**) Shannon, Simpson, ACE, and Chao indices reflecting alpha diversity. (**D**) Microbiota composition at the phylum level. (**E**) Histogram of LDA scores for taxonomic biomarkers identified by LEfSe. LDA scores (log 10) > 4 indicate enriched taxa in cases. Significance was determined using one-way ANOVA. *** represents *p* < 0.001.

**Figure 2 animals-14-00668-f002:**
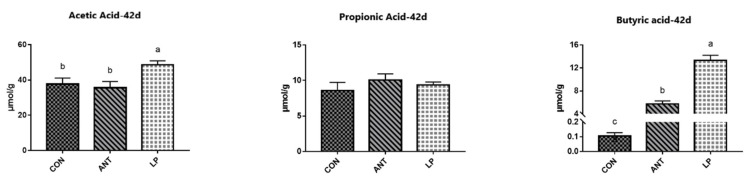
Effects of *Lactobacillus plantarum* on SCFAs in broilers. Values are presented as mean ± SEM. Individual broiler was regarded as the experimental unit. *n* = 8 per treatment. ^abc^ Means with different superscripts in the same row differ significantly at *p* < 0.05. Abbreviations: CON represents broilers fed a basal diet; ANT represents broilers fed a basal diet supplemented with 75 mg/kg chlortetracycline; LP represents broilers fed a basal diet and *Lactobacillus plantarumon*.

**Figure 3 animals-14-00668-f003:**
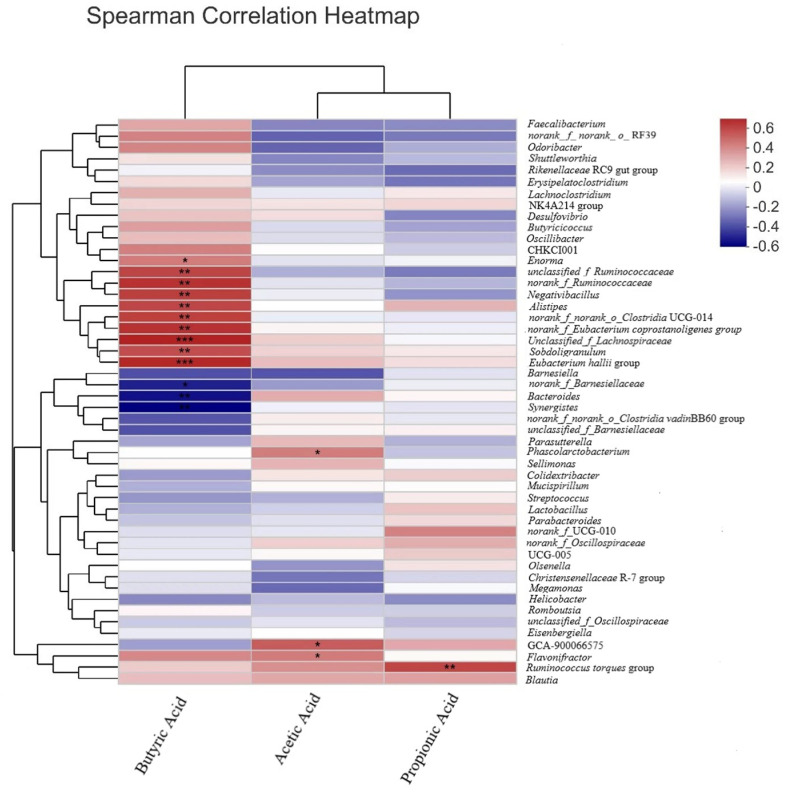
Correlation heatmap of the top 50 genera and SCFAs in broilers. Significance was determined using one-way ANOVA. * represents *p* < 0.05; ** represents *p* < 0.01; *** represents *p* < 0.001.

**Table 1 animals-14-00668-t001:** Ingredient composition and nutritional level of basic diet (air-dry basis).

Items	1–21 d	22–42 d
Ingredients (%)		
Corn	61.80	65.60
Soybean meal (CP, 43%)	22.50	17.55
Extruded soybean (CP,36%)	8.45	10.00
Fish meal (CP, 62%)	3.00	3.00
CaHPO_4_	1.66	1.45
Limestone	1.10	1.00
NaCl	0.32	0.30
DL-methionine	0.16	0.10
L-lysine	0.01	
Premix ^1^	1.00	1.00
Total	100.00	100.00
Nutrition levels (%)		
Metabolizable energy (MJ/kg)	12.45	12.70
Crude protein	21.00	19.20
Lysine	1.15	0.95
Methionine	0.54	0.44
Calcium	0.99	0.89
Available phosphorus	0.53	0.49

^1^ Premix is provided for feed per kg: VA, 1500 IU; VB_1_, 1.5 mg; VB_6_, 3.0 mg; VB_12_, 0.01 mg; VD_3_, 200 IU; VE, 10 IU; VK, 0.5 mg; biotin, 0.15 mg; D-pantothenic acid, 10 mg; folic acid, 0.5 mg; nicotinic acid, 30 mg; trace elements: Cu, Fe, Zn, Mn, Se, and I are 8 mg, 80 mg, 40 mg, 60 mg, 0.15 mg, and 0.18 mg, respectively.

**Table 2 animals-14-00668-t002:** Effects of dietary supplementation of *L. plantarum* on the growth performance of broilers.

Items	CON	ANT	LP	SEM	*p*-Value
BW, g					
1 d	38.29	39.52	38.32	0.401	0.361
21 d	732.08 ^b^	778.24 ^a^	800.13 ^a^	8.945	0.001
42 d	1875.86 ^b^	1926.47 ^b^	2124.36 ^a^	40.63	0.019
AWDG (g/d)					
1–21 d	33.54 ^b^	35.26 ^a^	36.31 ^a^	0.361	0.001
22–42 d	55.48 ^b^	56.47 ^b^	64.85 ^a^	1.764	0.048
1–42 d	44.51 ^b^	45.87 ^b^	52.57 ^a^	1.205	0.005
ADFI, g					
1–21 d	54.02 ^b^	54.42 ^ab^	59.84 ^a^	0.935	0.008
22–42 d	118.19	111.66	118.63	3.432	0.677
1–42 d	80.00	78.62	86.05	1.516	0.098
F:G					
1–21 d	1.61 ^ab^	1.54 ^b^	1.64 ^a^	0.017	0.036
22–42 d	2.13 ^a^	1.98 ^ab^	1.83 ^b^	0.046	0.023
1–42 d	1.80 ^a^	1.72 ^ab^	1.70 ^b^	0.018	0.067

^ab^ Means with different superscripts in the same row differ significantly at *p* < 0.05. Cage was regarded as the experimental unit. *n* = 8 per treatment. Abbreviations: CON represents broilers fed a basal diet; ANT represents broilers fed a basal diet supplemented with 75 mg/kg chlortetracycline; LP represents broilers fed a basal diet and *L. plantarumon*. F:G means feed–gain ratio.

**Table 3 animals-14-00668-t003:** Effects of *L. plantarum* on metabolite index in the serum of broilers.

Items	CON	ANT	LP	SEM	*p*-Value
NH_3_ (µmol/L)	10.83 ^a^	9.03 ^b^	6.54 ^c^	0.490	0.001
BUN (mmol/L)	0.31	0.29	0.23	0.016	0.136
UA (µmol/L)	147.84	129.88	116.41	6.584	0.148
XOD (U/L)	3.83 ^a^	4.30 ^a^	3.10 ^b^	0.168	0.005

^abc^ Means with different superscripts in the same row differ significantly at *p* < 0.05. Individual broiler was regarded as the experimental unit. *n* = 8 per treatment. Abbreviations: CON represents broilers fed a basal diet; ANT represents broilers fed a basal diet supplemented with 75 mg/kg chlortetracycline; LP represents broilers fed a basal diet and *Lactobacillus plantarumon*; BUN means blood urea nitrogen; UA means urea acid; XOD means xanthine oxidase.

**Table 4 animals-14-00668-t004:** Effects of *L. plantarum* on the antioxidant index of the serum of broilers.

Items	CON	ANT	LP	SEM	*p*-Value
GSH-PX (U/mL)	6.90	7.83	8.37	0.591	0.617
SOD (U/mL)	12.28 ^b^	12.96 ^ab^	13.22 ^a^	0.155	0.026
CAT (U/mL)	9.98 ^b^	10.79 ^ab^	11.80 ^a^	0.304	0.039
MDA (U/mL)	8.12	7.56	7.91	0.199	0.534

^ab^ Means with different superscripts in the same row differ significantly at *p* < 0.05. Individual broiler was regarded as the experimental unit. *n* = 8 per treatment. Abbreviations: CON represents broilers fed a basal diet; ANT represents broilers fed a basal diet supplemented with 75 mg/kg chlortetracycline; LP represents broilers fed a basal diet and *Lactobacillus plantarumon*; GSH-PX means glutathione peroxidase; SOD means superoxide dismutase; CAT means catalase; MDA means malonaldehyde.

**Table 5 animals-14-00668-t005:** Effects of *L. plantarum* on serum immunoglobulins and the immune response cytokines of broilers.

Items	CON	ANT	LP	SEM	*p*-Value
IgA ng/mL	168.11	177.93	176.81	4.327	0.626
IgY ng/mL	1.49 ^c^	1.79 ^b^	2.07 ^a^	0.067	0.001
IgM µg/mL	3.09 ^c^	3.61 ^b^	4.26 ^a^	0.143	0.001
IL-1β pg/mL	94.19 ^a^	83.02 ^b^	83.02 ^b^	1.900	0.011
IL-6 pg/mL	394.04	393.77	387.22	4.480	0.801
IL-10 pg/mL	17.43 ^b^	20.82 ^b^	28.43 ^a^	1.272	0.001
TNF-α pg/L	38.74 ^ab^	40.23 ^a^	37.35 ^b^	0.568	0.114
INF-β pg/mL	150.78	149.55	145.68	4.573	0.905

^abc^ Means with different superscripts in the same row differ significantly at *p* < 0.05. Individual broiler was regarded as the experimental unit. *n* = 8 per treatment. Abbreviations: CON represents broilers fed a basal diet; ANT represents broilers fed a basal diet supplemented with 75 mg/kg chlortetracycline; LP represents broilers fed a basal diet and *Lactobacillus plantarumon*.

## Data Availability

The data that support the findings of this study are available from the corresponding author upon reasonable request.

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
