# Peer review of "Effects of Probiotic Lactiplantibacillus plantarum HJLP-1 on Growth Performance, Selected Antioxidant Capacity, Immune Function Indices in the Serum, and Cecal Microbiota in Broiler Chicken"

_animals, 2024, doi:10.3390/ani14050668_

Round 1
Reviewer 1 Report
Comments and Suggestions for Authors
Comments to the Authors of manuscript number: animals-2779366 entitled “Effects of probiotic Lactoplantibacillus plantarum on growth performance, antioxidant capacity, immune function, and cecal microbiota in broiler chicken”.
The research investigated the effects of L. plantarum on broiler growth, oxidation resistance, immunity, and cecal microbiota. Broilers were divided into control, antibiotic (chlortetracycline), and probiotic (L. plantarum) groups. L. plantarum supplementation improved daily weight gain, reduced feed-to-gain ratio, lowered serum ammonia and xanthine oxidase levels, and enhanced antioxidant enzyme and antibody levels. L. plantarum also influenced the balance of pro-inflammatory and anti-inflammatory factors. Moreover, it increased concentrations of beneficial short-chain fatty acids in cecal contents and positively impacted cecal microbiota composition. The study suggests L. plantarum as a promising antibiotic alternative in poultry breeding.
1. All kits used in the study should be described by the catalog number, the detectable minimum.
2. L 151- why serum ammonia was detected. It should be explained
3. the hypothesis is presented
4. the specific goal of the study is well-established
5. the study design is presented well. Proper number of birds was used
6. the discussion is comprehensive and logically performed
7. Specify if the weights of broilers were taken individually or as a group in each pen.
8. Detail any specific procedures for aseptically removing cecal contents.
9. Provide a list of the specific biochemical parameters analyzed in serum samples. Mention the units of measurement for each parameter. Include any quality control measures undertaken during the analysis.
10. Specify the specific diagnostic kits used for assessing microbial genomic DNA.
Provide more details on the alpha and beta diversity analyses, including the software/tools used.
11. Clarify the rationale behind choosing the 338F-806R region for 16S rRNA sequencing.
Reviewer 2 Report
Comments and Suggestions for Authors
Dear Corresponding Authors,
Please find the specific comments and suggestions listed line-by-line below.
Title - Please consider changing the title to "Effect of Lactiplantibacillus plantarum on the growth performance, selected antioxidant capacity, immune function indices in the serum, and cecal microbiota in broiler chickens." After L. plantarum, the DSM number must be added in the brackets.
Simple summary
L15 - L. plantarum should be expanded.
L17 - broiler nutrition instead of feeding
L18-19 - "on the growth performance, oxidation resistance, immune response, intestinal microbiota, and short chain fatty acid profiles." Please be more precise. Oxidation resistance and immune response, but where exactly? intestinal microbiota or cecal microbiota? SCFA profile, but where?
Abstract
L21-22 - find the comment above, and be precise.
L23 - there is a lack of information about the birds' sex.
L41 - cecal microbiota instead of gut microbiota
L42 -tremendous - please delete
Keywords - please do not use the exact words from the title
Introduction
L57 - have been banned in China
L60 - diets instead of rations
L61 - microbiota instead of flora
L64 - "aerotolerant" or aerobic?
L66-67 - This is an unnecessary sentence; please delete it.
L69-70 - please do not cite articles about mice and rats. In the introduction section, the authors should present the latest knowledge about the usage of L. plantarum based on the research conducted on poultry.
L75-78 - the authors must be precise; please see comments above.
Comment 1 - the introduction section is poorly written. There is a need to add the latest information about the effect of L. plantarum on growth performance, selected immune traits, oxidation, and cecal microbiota. Please find similar papers and show the necessity of continuing this topic. Additionally, the hypothesis addition is needed.
Material and methods
Comment 2 - at the very beginning of this section the information about the local ethical committee is obligatory.
L81 - what does the Avian chickens mean? What about the sex of birds?
L84 - why was the L. plantarum concentration established at the level off 5*10^8 CFU/kg?
L87 - why do the authors use old (1994) nutrient requirements for poultry? Usually, the hybrid producers share the nutrient specifications for each hybrid.
L88-89 - the environmental conditions should be significantly expanded.
Table 1 - soybean meal - please add the CP level, also for extruded soybean and fish meal (is it essential that fish meal was imported?). If the nutrition levels were calculated, please also change to "chemical composition" in the table title.
Comment 3 - No information about the diet preparation (technical point of view), other feed additives supplementation such as coccidiostats, carbohydrates, phytase, etc., and vaccine program were highlighted. Additionally, there is no information about the experimental factor preparation., as well as about the physical form of the product.
L100 - "were randomized", meaning one bird from each replicate pen was randomly chosen. If yes, please correct.
L103 - Cecal content cannot be aseptically removed; it is impossible to make it like that. The reviewer suggested removing this sentence.
L105 - this subsection is poorly written. The material and methods section should be presented to allow future readers to repeat this study. Please add information about the equipment (device name, company name, city, country), chemicals, and kits (chemical/kit name, company name, city, country). Please be precise.
L114 - microflora is an ancient term in the case of 16S analyses. The full explanation of the alpha and beta diversity analyses should also be presented.
Comment 4 - The subheading, i.e., 2.5, is not nearly written enough. Please find the example of how to describe this kind of analysis.
Trela, J., KieroÅ„czyk, B., Hautekiet, V., & Józefiak, D. (2020). Combination of Bacillus licheniformis and salinomycin: Effect on the growth performance and GIT microbial populations of broiler chickens. Animals, 10(5), 889. https://doi.org/10.3390/ani10050889
Comment 5 - The manuscript regarding the Statistical Analysis subsection should be improved before further processing. Please find the specific comment below,
There is no information about testing normal distribution and variance homogeneity. Thus, it is not possible to judge the correctness of ANOVA usage. There is a high possibility that raw data must be transformed (due to the lack of normal distribution occurrence), or the Welsh ANOVA should be done when the homogeneity of variance is significant. This information is essential to add. Furthermore, in this section, there is a lack of information on what was defined as an experimental unit in terms of each parameter. Based on the reviewer's experience, it is rare to obtain normally distributed physiological and microbiological data. Thus, this issue should be double-checked. In the response, please add a separate file with a table including the p-value from the Shapiro-Wilk and Bartlett tests.
The full description of how the Venn graph, principal component analyses, alpha diversity, and LEfSe should be added. Spearman correlation should also be described in terms of its strength.
Due to the material and methods section issues, the reviewer must stop the review process at this stage.
Kind regards,
Section Board Editor
Author Response
Responses to reviewer 2
Thank you so much for your suggestions. We have carefully revised the text in accordance with your advice, which our responses are as followed
1.Title - Please consider changing the title to "Effect of Lactiplantibacillus plantarum on the growth performance, selected antioxidant capacity, immune function indices in the serum, and cecal microbiota in broiler chickens." After L. plantarum, the DSM number must be added in the brackets.
Response: Modified
- Simple summary
L15 - L. plantarum should be expanded.
Response: Modified
L17 - broiler nutrition instead of feeding
Response: Modified
L18-19 - "on the growth performance, oxidation resistance, immune response, intestinal microbiota, and short chain fatty acid profiles." Please be more precise. Oxidation resistance and immune response, but where exactly? intestinal microbiota or cecal microbiota? SCFA profile, but where?
Response: Modified
3.Abstract
L21-22 - find the comment above, and be precise.
Response: Modified
L23 - there is a lack of information about the birds' sex.
Response: we’ve added the birds’ sex information in L80.
L41 - cecal microbiota instead of gut microbiota
Response: Modified
L42 -tremendous - please delete
Response: Deleted
Keywords - please do not use the exact words from the title
Response: Modified
- Introduction
L57 - have been banned in China
Response: Modified
L60 - diets instead of rations
Response: Modified
L61 - microbiota instead of flora
Response: Modified
L64 - "aerotolerant" or aerobic?
Response: anaerobic-aerotolerant
L66-67 - This is an unnecessary sentence; please delete it.
Response: Deleted
L69-70 - please do not cite articles about mice and rats. In the introduction section, the authors should present the latest knowledge about the usage of L. plantarum based on the research conducted on poultry.
Response: This also is an unnecessary sentence, we’ve deleted it.
L75-78 - the authors must be precise; please see comments above.
Comment 1 - the introduction section is poorly written. There is a need to add the latest information about the effect of L. plantarum on growth performance, selected immune traits, oxidation, and cecal microbiota. Please find similar papers and show the necessity of continuing this topic. Additionally, the hypothesis addition is needed.
Response: Modified
In the light of the information presented above, we evaluated L. plantarum to modulate growth and health in broiler chickens compared with antibiotic growth promoters to establish whether L. plantarum plays an efficient role in replacing antibiotics as growth promoter in broiler feeding. To complete the objective mentioned, in addition to growth performants, several parameters were studied such as the antioxidant concentration, immune function indices, cecal microbiota, and some short chain fatty acids as those are the key factors to determine the health status in the birds.
5.Material and methods
Comment 2 - at the very beginning of this section the information about the local ethical committee is obligatory.
L81 - what does the Avian chickens mean? What about the sex of birds?
Response: It’s Avian broiler, we’ve modified. The sex of the birds is male, we’ve added in L78.
L84 - why was the L. plantarum concentration established at the level off 5*10^8 CFU/kg?
Response: This concentration is the optimal amount of addition according to the results of the previous broiler breeding experiment.
L87 - why do the authors use old (1994) nutrient requirements for poultry? Usually, the hybrid producers share the nutrient specifications for each hybrid.
Response: This animal experiment was carried out in August 2020 at a chicken farm and the feed and its formula was supplied by the farm.
L88-89 - the environmental conditions should be significantly expanded.
Response: Modified
Table 1 - soybean meal - please add the CP level, also for extruded soybean and fish meal (is it essential that fish meal was imported?). If the nutrition levels were calculated, please also change to "chemical composition" in the table title.
Response: Modified
Comment 3 - No information about the diet preparation (technical point of view), other feed additives supplementation such as coccidiostats, carbohydrates, phytase, etc., and vaccine program were highlighted. Additionally, there is no information about the experimental factor preparation., as well as about the physical form of the product.
L100 - "were randomized", meaning one bird from each replicate pen was randomly chosen. If yes, please correct.
Response: Corrected
L103 - Cecal content cannot be aseptically removed; it is impossible to make it like that. The reviewer suggested removing this sentence.
Response: modified
L105 - this subsection is poorly written. The material and methods section should be presented to allow future readers to repeat this study. Please add information about the equipment (device name, company name, city, country), chemicals, and kits (chemical/kit name, company name, city, country). Please be precise.
L114 - microflora is an ancient term in the case of 16S analyses. The full explanation of the alpha and beta diversity analyses should also be presented.
Response: details on the alpha and beta diversity analyses, including the software/tools used were provided in L137-151.
Comment 4 - The subheading, i.e., 2.5, is not nearly written enough. Please find the example of how to describe this kind of analysis.
Response: Modified
Comment 5 - The manuscript regarding the Statistical Analysis subsection should be improved before further processing. Please find the specific comment below,
There is no information about testing normal distribution and variance homogeneity. Thus, it is not possible to judge the correctness of ANOVA usage. There is a high possibility that raw data must be transformed (due to the lack of normal distribution occurrence), or the Welsh ANOVA should be done when the homogeneity of variance is significant. This information is essential to add. Furthermore, in this section, there is a lack of information on what was defined as an experimental unit in terms of each parameter. Based on the reviewer's experience, it is rare to obtain normally distributed physiological and microbiological data. Thus, this issue should be double-checked. In the response, please add a separate file with a table including the p-value from the Shapiro-Wilk and Bartlett tests.
Response: The experimental data were normally distributed and tested for homogeneity of variance. For example, data results for NH3 in serum. A, B, C in the graph represents the normal distribution test for CON(A), ANT(B), LP(C) respectively.
The full description of how the Venn graph, principal component analyses, alpha diversity, and LEfSe should be added. Spearman correlation should also be described in terms of its strength.
Response: described in L142-151.
Round 2
Reviewer 2 Report
Comments and Suggestions for Authors
L24 - "three hundred and sixty 1-day-old broilers" add sex
Comment 1 - why did the authors add exactly 5×10^8 CFU kg-1 of Lactoplantibacillus plantarum? What was the reason?
L35 - "dramatically" delete
L36 - "dramatically" delete
L39 - Latin names of bacteria populations should be written without the italic font. Only genus and species must be written using italic font. Please correct the whole manuscript accordingly.
Comment 2 - The abstract section length should be suitable for authors' instructions.
Keywords - please do not repeat terms from the title. Choose alternatives.
Introduction
Comment 3 - the introduction section is written poorly. The most essential information is avoided. Please explain why the authors decided to examine birds' growth performance, antioxidant capacity, immune function indices, and cecal microbiota. What is an L. plantarum mode of action? What about data published to date? What was the hypothesis of the presented study? From the reviewer's point of view, the Introduction section is too short.
Material and methods
L78 - what does "avian broilers" mean? Please mention the exact hybrid name.
L84- why the authors used old nutrient requirements, i.e., published in 1994. All hybrid producers frequently published updated version for each birds breeds.
Comment 4 - there is no information about the environment, except the temperature. What about the humidity, lightning programme, litter, vaccination, the number of feeders, and watering line/nipples, etc.
Comment 5 - no information about the diet preparation, there is only info that experimental diets were provided in the mash form, however, no information about thee reason of this decision is appeared in the text.
Comment 6 - The growth performance and sample collections subsections should be connected.
L101-103 - please add the centrifuge name, company name, city and country
L147-148 - "Venn diagrams were also generated to compare the numbers of common and unique OTUs among three groups." please add the specific package which was used to generate venn diagramm
L151 - "psych package" - it is necessary to add version.
L160 - There is no information about testing normal distribution and variance homogeneity. Thus, there is not possible to judge the correctness of ANOVA usage. There is a high possibility that raw data must be transformed (due to the lack of normal distribution occurrence), or the Welsh ANOVA should be done when the homogeneity of variance is significant. This information is essential to add.
Furthermore, in the presented study, the correlation was used; however, no information about which correlation, i.e., spearman, person, no strength definitions, was presented as well.
Results
L170 - "during the start period (day 1 to 21)"; the reviewer does not understand... in Table 2 the authors mentioned results from 21st day, not a period form 1-21d... please, correct
170-171 - the same as above. Only result from 42d is mentioned for BW, not for the perion 22-41d... please correct.
L170,172,175 - starter period instead of start period.
L173 - "ADWG" there is no abbreviation in the Table 2... please correct
L175-176 - it is not true, control group did not differ in te starter period in terms of F/G
Comment 7 - the growth performance in the results section should be rewise and rewritten. No information about the feed intake results. The shortcuts should be mirrored those used in Tables.
Comment 8 - the Results section should present the exact p-values in the text, not only p < or > 0.05. It is not enought.
L190,192,228 - editing
L228 - please do not use microflora term, use microbiota instead in the whole manuscript.
Figure 2 - propionic not propanoic acid
Discussion
L361 - Alistipes
Comment 9 - the discussion part can be improved including the proposition of the mode of action of the experimental factor, particularly in microbiological part. Changes in microbiota is poorly presented and explained.
L370 - dramatic - delete
